# Effects of Polymer–Curing Agent Ratio on Rheological, Mechanical Properties and Chemical Characterization of Epoxy-Modified Cement Composite Grouting Materials

**DOI:** 10.3390/polym16182665

**Published:** 2024-09-22

**Authors:** Yuxuan Wang, Jiehao Wu

**Affiliations:** 1School of Civil Engineering and Architecture, Anhui University of Science and Technology, Huainan 232001, China; wangyx95@foxmail.com; 2Zhiruiyuan Traffic Consultation Limited Company, Chongqing 400030, China

**Keywords:** polymer-modified cement, water-borne epoxy resin, grouting material, rheological property

## Abstract

This study designs and uses water-borne epoxy resin (WBER) and curing agent (CA) to modify traditional cement-based grouting for tunnels. The purpose of this paper is to analyze the rheological and mechanical properties of composite grouting with different ratios of WBER and CA and analyze the modification mechanism by means of chemical characterization to explore the feasibility of WBER as a high-performance modifier for tunnel construction. The composite grouting is prepared by mixing cement paste with polymer emulsion. A series of experiments was carried out to investigate the effects of WBER and CA, including the slump test, viscosity, rheological curve, setting time, bleeding rate, grain size distribution, zeta potential, compressive and splitting tensile strength, X-ray diffraction(XRD), Fourier-transform infrared spectroscopy (FT-IR), and scanning electron microscopy (SEM), on the composite grout. The results show that WBER improves grout fluidity, which decreases in combination with CA, while also reducing the average particle size of the composite grout for a more rational size distribution. Optimal uniaxial (38.9%) and splitting tensile strength (48.7%) of the grout are achieved with a WBER to CA mass ratio of 2:1. WBER accelerates cement hydration, with the modification centered on the reaction between free Ca^2+^ and polymer-OH, significantly enhancing the strength, fluidity, and stability of the polymer-modified composite grout compared to traditional cement-based grouting.

## 1. Introduction

In tunnel engineering, it is imperative not only to improve construction technology to address the array of complex geological challenges but also to continuously safeguard against the potential risks of underground subsidence and water seepage within tunnels. [1] The shield tunneling method is widely used in underground construction due to its low labor cost, high construction efficiency, safety, reliability, and less disturbance to the aboveground environment [2,3,4]. Concurrently, the grouting technology used in conjunction with shield tunneling technology has a very prominent effect in solving ground subsidence and tunnel water seepage. The hardened grouting filled behind the shield lining can not only stabilize the stratum but also act as a protective wall to prevent groundwater and erosion media from entering the tunnel, effectively reducing the risk caused by the surrounding rock deformation to construction [5,6].

Neat cement paste is not suitable for grouting in shield tunnels due to its inherent performance deficiencies. There are three kinds of commonly used modified additives: (a) industrial waste, such as fly ash [7], slag [8], steel slag [9], and silica fume [10]; (b) special soil, such as bentonite [11], gypsum [12], clay, and metakaolin [13]; and (c) chemical additives, such as superplasticizers [14], plastic expansive agents [15], and viscosity modifying agents [16]. It is a new research direction to use a polymer to modify cement paste. The addition of a polymer can effectively improve the performance of cement-based grouting. The polymer forms a film inside the cement paste, which is used to restrain cracking caused by shrinkage of the cement paste [17], improve the waterproof property [18], and improve the tensile strength and flexural strength. At present, the research on polymer-modified cement in grouting engineering practices mainly includes tunnel support reinforcement [19,20], pavement repair [21,22], soil reinforcement [23,24,25], and cementing fluid in petroleum engineering [26].

The high adhesion and chemical resistance of epoxy resin (ER) have led to its widespread utilization as both an adhesive and a corrosion-resistant coating in the construction industry. ER is a prevalent chemical grouting, which is usually used in foundation reinforcement, road maintenance, and tunnel support. ER, with active groups such as epoxy groups, also plays a certain role when used as a cement admixture [27]. The water-borne epoxy resin (WBER) employed in this study is an aqueous emulsion liquid, which is synthesized by the chemical synthesis of bisphenol-A diglycidyl ether-type ER and compounds containing hydrophilic groups. ER, especially WBER, has great potential in the production of polymer-modified concrete due to its advantages such as improving the workability and mechanical properties of concrete, enhancing volume stability, and improving resistance to chloride ion penetration and chemical corrosion [28,29]. ER is cured with cement to form a crosslinked polymer network in the internal pores of cement, reduce the porosity of composites, and improve the elastic modulus and flexural properties to a certain extent [30,31]. ER can notably improve the bonding capacity of the composite grout [32,33], increase the stability of the fresh paste, and at the same time, reduce the bleeding ratio. Additionally, the good compatibility between epoxy resin and fiber can be used to prepare fiber-reinforced composite materials [34].

The traditional view is that ER should be used together with a curing agent (CA) when modifying cement. However, under tunnel construction conditions, the combined use of ER and CA will make the preparation of composite grouting more complicated. Ohama [35] found that ER can harden under the catalysis of OH- produced in the weak alkaline environment of cement, so hardener-free epoxy-modified mortar (HFEMM) without CA was successfully developed. Figure 1 shows the hardening mechanism of ER without CA inside HFEMM. Bhutta [36] used HFEMM to modify mortar panels. Epoxy-modified mortar panels have stronger ductility and higher bearing capacity than unmodified mortar panels. Nur farhayu ariffin [37] used the wet–dry curing method for HFEMM, and the compressive, flexural, and tensile strength of HFEMM were improved compared with unmodified paste. Single-component modifications have more advantages than modifying two components in engineering practice.

Although WBER-modified cement mortar has been studied and applied in civil engineering, its application in grouting engineering has not been well explored. With the rapid development of deep underground engineering, the formation, prediction, and control of geological disasters such as fracture, swelling, deformation, and permeability of deep surrounding rock are becoming increasingly prominent. Traditional cement-based grouting materials have poor bonding performance and brittle failure characteristics, which make it difficult to meet the reinforcement needs of nonlinear large deformation and large block displacement. Therefor, a new ER-modified composite grouting was prepared using cement paste and polymer liquid, which is composed of WBER, CA, and defoamer with different ratios. The rheological and mechanical properties of the composite grouting were studied through a series of experiments, including fluidity, setting time, bleeding, viscosity, rheological curve, grain size distribution, zeta potential, uniaxial compressive strength, and splitting tensile strength. The composition, crystal form, and functional groups of the hydration products of grouting were studied by XRD and FT-IR. The micro-morphology of the composite grouting was observed by SEM, and the modification mechanism of WBER to neat cement paste was discussed.

## 2. Materials and Methods

### 2.1. Materials

In the field of grouting technology, it is common practice to utilize a cement paste with an elevated water-to-cement (w/c) ratio to guarantee the necessary fluidity and to enhance the effectiveness of the grouting process. Cement paste was prepared with ordinary Portland cement (OPC, P·O 32.5) manufactured by Huaxin Company. WBER (molecular formula: C_21_H_24_O_4_) is bisphenol-A diglycidyl ether-type ER and the number of epoxy groups in WBER is two. The corresponding CA (molecular formula: C_12_H_12_O_2_N_2_S) used in this paper was 4′4-sulfonyldianiline. Figure 2 illustrates the chemical reaction equation for WBER and CA. WBER and CA were both produced by Shenzhen Jitian Chemical Co., Ltd. The mixture of WBER and CA was only calculated based on the quality of the active solid components when preparing the sample. Table 1 displays the characteristics of WBER and CA. Defoamer (tributyl phosphate, TBP) was added to the composite paste to deal with a large number of bubbles generated after the addition of WBER. CK-6206-type defoamer was produced by Tianjin Gaotian New Material Technology Co., Ltd. The amount of the defoamer added was 1% of the polymer content of WBER. Table 2 outlines the properties of the defoamer. Deionized water was used in XRD, FT-IR, and CLSM. In addition, urban tap water was used in other experiments, and the water contained in the polymer should be deducted when calculating the actual water used for grouting.

### 2.2. Preparation of WBER-Modified Cement-Based Paste

The ambient temperature for sample preparation was room temperature (25 ± 2 °C), the water–cement ratio was 0.6, and the mass percentage of polymer to cement was 10%. The control group consisted of neat cement paste. The preparation process of the WBER-modified cement composite grouting material is shown in Figure 3. WBER and CA in different proportions were added to the beaker. The mixture was stirred evenly with an electric stirrer for two minutes at a rate of 200 rpm. Then, the defoamer was added, stirring for 1 min, and the polymer emulsion was prepared. Ordinary Portland cement and water were added into the mortar mixer and mixed at a speed of 200 rpm for 2 min until the cement and water were evenly mixed. Finally, the cement paste was combined with the emulsion and agitated for a duration of 5 min at 200 rpm, thereby ensuring uniform distribution of the polymer throughout the unmodified cement paste.

### 2.3. Experimental Approaches

#### 2.3.1. Fluidity and Rheological Properties

The slump test, which serves as both a method and an indicator for assessing concrete, is specifically designed to evaluate the fluidity of mixed concrete. Conducting a slump test using a metal container in the shape of a truncated cone, with the following dimensions: a top base radius of 3.6 cm, a bottom base radius of 6 cm, and a total height of 6 cm. The rheological properties of fresh paste were tested using a rotational viscometer (instrument model: ZNN-D6B). The viscometer has six speeds, which range from 3 to 600 rpm, the shear rate increased in the range of 0 to 1022 s^−1^, and the speed of each stage was maintained for 20 s. The rheological curve of the composite grout was drawn according to the relationship between the shear rate and the shear stress. The instrument is capable of displaying the torque applied to the spindle surface, and the shear stress and apparent viscosity can be calculated using Equations (1) and (2), as follows:(1)$$\tau =\frac{M}{2\pi R^{2}L}$$
(2)$$\eta =\frac{\tau }{\upsilon }$$
where *τ* = shear stress, *η* = apparent viscosity, *M* = torque read on the instrument, *R* = radius of the inner spindle, *L* = length of the spindle, and *υ* = shear rate.

#### 2.3.2. Grain Size Distribution and Zeta Potential

The grain size distribution with a range of 0.04 μm~2000 μm of fresh paste was analyzed using a laser particle size analyzer (instrument model: Bettersize2000E, produced by Bettersize Instruments Ltd.). The sample was dispersed by ultrasound to ensure uniform dispersion of particles, and deionized water was used as the dispersant. The zeta potential of fresh paste was measured using a zeta potential analyzer (instrument model: Nano ZS90, manufactured by Malvern Panalytical Instruments Ltd.). The sample was kept mixed as evenly as possible to ensure that the electrical double layer (EDL) reached equilibrium before the measurement.

#### 2.3.3. Bleeding Ratio and Setting Time

With reference to GB/T 1346-2011 [39], the setting time of fresh polymer-modified paste was measured using the Vicat apparatus with the penetration depth as the index. Bleeding is the phenomenon of coarse aggregate sinking and water floating during concrete transportation, vibration, and pumping, which is a significant aspect of the workability of freshly mixed paste. About 100 mL of freshly mixed paste was weighed in a measuring cylinder and sealed with plastic film. The volume of the solid part in the measuring cylinder was read every 30 min until the solid part no longer decreased after 6 h. The bleeding ratio can be determined by Equation (3), as follows:(3)$$Bleeding\;ratio=\frac{V_{t}-V_{s}}{V_{t}}$$
where *V_t_* = total volume of initial paste (mL), and *V_s_* = volume of remaining solids (mL).

#### 2.3.4. X-ray Diffraction (XRD) and Fourier-Transform Infrared Spectroscopy (FT-IR)

The crystalline state and phase composition of the polymer-modified grout could be determined by an X-ray diffraction experiment, and the dried powder samples were analyzed and tested using an X-ray diffractometer (instrument model: XPert Pro) from Malvern Panalytical. A Fourier-transform infrared spectrometer (instrument model: Nicolet is 5) manufactured by Thermo Fisher Scientific, Inc. was used to detect the infrared spectrum of hardened grout samples in the wavenumber range of 400–4000 cm^−1^. Approximately 300 mg of potassium bromide powder was mixed with each milligram of sample powder, and the sample was dried and then prepared using the pressed-disk technique. The experimental group in XRD and FT-IR was WBER/CA = 2:1, and the samples were cured for 28 d.

#### 2.3.5. Scanning Electron Microscopy (SEM) and Confocal Laser Scanning Microscopy (CLSM)

A scanning electron microscope (instrument model: S-3400N) produced by Hitachi was used to capture the SEM images of the grout samples that were cured for 28 days. Secondary electron imaging was used for scanning electron microscopy, and the acceleration voltage and emission current were set at 20 kV and 70 μA, respectively. Due to the poor conductivity of cement-based materials, it is necessary to spray gold on the sample surface to increase its conductivity before the experiment. The distribution and morphology of WBER in fresh cement paste were observed by CLSM (instrument model: LSM800, produced by Carl Zeiss). The sample was selectively labeled with a dye, and the labeled object presented the specified color under the excitation of a specific wavelength laser so that the composition and distribution characteristics of the sample could be distinguished simply by color. The polymer phase in the composite grouting was stained and labeled with the lipophilic dye Nile red, which was red under excitation by a 488 nm laser. By contrast, the aqueous phase was labeled with sodium fluorescein, which showed green under a 561 nm laser. At 48 h before the experiment, the polymer emulsion was dyed with Nile red dye at a concentration of 100 ppm to ensure uniform staining of the polymer. Prior to preparing the sample, deionized water was mixed with sodium fluorescein at a concentration of 100 ppm to label the aqueous phase within the freshly mixed paste. During the experiment, the fresh paste was dropped on the slide and sealed with a cover glass to prevent water evaporation. The micromorphology of the sample was observed with 20 times and 40 times oil immersion objective lenses.

#### 2.3.6. Mechanical Properties

The uniaxial compressive strength and Brazilian disc splitting tests on the samples were conducted using the Rmt-301 rock and concrete testing system. The uniaxial compressive test used a 50 mm × 100 mm cylinder mold (refers to ASTM C942 [40]) with a loading rate of force of 0.05 kN/s. The Brazilian splitting experiment used a 50 mm × 50 mm cylinder mold to prepare the sample, and the loading rate of the force during the experiment was 0.05 kN/s.

## 3. Results and Discussion

### 3.1. Fluidity and Rheological Properties of Epoxy-Modified Composite Grouting

Figure 4 illustrates the influence of the polymer on the initial fluidity of the altered composite paste. As is evident in Figure 4, both WBER and CA exerted a pronounced impact on the fluidity of the composite grout. The fluidity of the composite grouting increased from 258.9 mm to 288.5 mm when only pure WBER was added. However, with the increase in CA content, the fluidity of the composite grouting decreased. This indicated that the cement paste with WBER had a dispersion effect on the fresh suspension system, which was attributed to the geometric differences between WBER and cement particles. The hydration products of cement particles are usually irregular and amorphous, while the particles of WBER are spherical. The spherical WBER particles can have a “ball bearing” effect in the hydration products of cement, thereby reducing the surface friction between the irregular particles of cement hydration products, which in turn enhances the fluidity of the grouting mixture [41]. WBER reacts with CA to form a longer polymer chain, which reduces the fluidity of the composite grouting. Figure 4 also records the change in the fluidity of the composite grouting within 3 h after adding water. It can be seen from the curve that, with the extension of mixing time, the hydration reaction inside the suspension system continues to generate a flocculation structure, and the fluidity of the grouting continues to decline. The fluidity of the control group decreased from 258.9 mm to 130.65 mm, with a decrease of 49.5%, while that of the other groups decreased by 33% at the maximum. This indicated that WBER can prolong the flowable time of fresh composite paste, which makes the grout maintain good pumpability, improves its workability for a longer period of time, and improves the practicability of grout at the construction site.

#### 3.1.1. Rheological Curve

Figure 5 depicts the rheological curves of composite pastes with varying ratios of WBER to CA, as measured using a rotational viscometer. This device was employed to quantify the shear stress of the modified composite paste containing different polymer concentrations at distinct shear rates, with the resulting rheological data presented in Figure 5. Cement-based grouting usually behaves as a non-Newtonian fluid, and on most occasions, the rheological curve of cement-based grouting tallies with the Bingham model, which is relatively simple, easy to use, and widely used in the engineering field [42,43]. The Bingham model was used to fit the rheological curve of the polymer-modified cement-based grout. The fitting parameters, namely, yield stress *τ*_0_ (the intercept of the axis of shear stress), plastic viscosity *η* (the slope of the fitting line), and the correlation coefficient *R*^2^, are shown in Table 3. According to the correlation coefficient *R*^2^, the fitting results were good, which indicated that the rheological experimental data were in good agreement with the characteristics of a Bingham fluid.

The yield stress is an important index to measure the initial flow of grout, and the flow of paste must satisfy the condition that shear stress exceeds its yield stress [30]. The greater the yield stress, the more difficult for grout to flow. During the grouting process, the yield stress of grout is the main resistance of pumping, and appropriate yield stress is crucial to guarantee seamless pumping and effective diffusion of the paste. As can be seen from Figure 5, the yield stress of grouting decreased with the addition of polymer, decreasing first and then increasing with the increase in the proportion of CA in the polymer. The reason may be that a small amount of WBER adsorbs on the surface of cement particles, inhibiting the early hydration cementitious bonding of cement and preventing the close network structure from forming inside the paste, and WBER also plays a lubricating role between cement particles. The plastic viscosity has a significant influence on the continuity and homogeneity of fresh grout as well as on the mixing and bonding properties between pastes. A low plastic viscosity of fresh grout will lead to the adverse effects of extrusion and bleeding in the process of pumping and diffusion, which will lead to the blockage of pipes and affect the pumping of grout. On the flip side, when the plastic viscosity is small, the deformation resistance after grouting is low and the plastic deformation is large. Modification with WBER reduced the plastic viscosity of epoxy-modified paste and improved the fluidity of the grout. The plastic viscosity of WBER-modified composite paste increased with the increase in the amount of CA. The use of CA can improve the adhesive property of the grout and make the plastic viscosity of the paste adjustable within a certain range.

#### 3.1.2. Zeta Potential and Grain Size Distribution

In the investigation of grouting materials, the grain size distribution within the water-mixed suspension is of paramount importance for determining the fluidity and spreadability of grout, overshadowing the importance of the raw particle size. Utilizing a laser particle size analyzer, we conducted measurements of the particle size distributions for both unmodified and epoxy-modified cement pastes, as presented in Figure 6. Figure 6a shows that the incorporation of WBER significantly decreased the average particle size of the composite paste, with a significant increase in the proportion of particles smaller than 10 μm, from 30.7% to 63.4%. According to the literature [42], the particle size distribution of grout serves as an effective indicator for evaluating its permeability and operational performance. In Figure 6b, the addition of WBER resulted in a decrease in the distribution curve within the 10–100 μm range, while concurrently, there was an upward shift in the curve for particle sizes below 10 μm. This indicated that WBER modified the particle size distribution of the paste, reducing the suspension’s content of larger particles and enhancing the proportion of smaller particles. Given that the particle size of WBER is finer than that of cement, its incorporation into the grout aided in refining the grain size distribution and decreasing the overall particle size of the grout, thereby improving its working performance.

Zeta potential is a critical indicator of the surface charge status of particles and is pivotal in assessing the fluidity of grouting materials. It determines the intensity of the electrostatic interactions among particles, which directly influences the rheological behavior of suspensions [44]. A higher absolute zeta potential value correlates with a more significant dispersion of particles within the suspension.

Figure 7 illustrates how the zeta potential of WBER-modified composite grout was affected by the ratio of WBER to CA. Following the addition of water, the cement’s water-soluble constituents dissolved rapidly, leading to an observed zeta potential of +3.74 eV for the control sample (unmodified cement paste). This is attributed to the swift hydration reactions of calcium-containing compounds such as C_3_S, C_2_S, and C_3_A, which release a multitude of calcium ions. As reported in the literature [45], the dissolution and subsequent adsorption of calcium ions in cement suspensions result in the formation of positive charge sites on particle surfaces, thereby elevating the zeta potential. The introduction of WBER to the mixture caused a rise in the zeta potential of the cement suspension. However, with the rising proportion of CA, the zeta potential underwent a decrease and ultimately became negative. This indicated that WBER improved the grout’s flowability, with the optimal flow achieved at a lower CA content. These results aligned with previous assessments of the fluidity and rheological characteristics.

#### 3.1.3. Setting Time and Bleeding Ratio

Figure 8 shows the influence of different WBER and CA ratios on the setting time of WBER-modified composite paste. As observed in Figure 8, with the addition and the change in the ratio of WBER and CA, the setting time of composite grout did not change significantly. It showed that the setting time was not affected when less polymer was added (10%).

Figure 9 demonstrates the effect of varying WBER and CA ratios on the bleeding ratio of the composite grouting. As depicted in Figure 9, the addition of WBER markedly diminished the bleeding ratio of the composite grout. At a WBER to CA ratio of 2:1, the 6 h bleeding ratio was reduced to 1.37%, compared to 3.89% in the control group, representing only 35.2% of the control group’s bleeding ratio. The bleeding ratio was further decreased in grouting samples containing CA, relative to those without CA, with no significant differences observed among experimental groups with different CA ratios. WBER induced the formation of a composite multilayer film structure within the altered paste, effectively entrapping the free water. As indicated by the zeta potential measurements, the addition of WBER decreased the absolute value of the zeta potential in the suspensions, which reduced the repulsion between cement particles, enhanced the flocculation structure, and consequently diminished the amount of free water released. Thus, the addition of WBER improved the stability and reduced the bleeding ratio of the grouting mixture.

### 3.2. Chemical Characterization of Epoxy-Modified Composite Paste

#### 3.2.1. Fourier-Transform Infrared Spectroscopy (FT-IR)

The FT-IR spectrum of WBER-modified composite material is shown in Figure 10. The FT-IR spectrum was analyzed by observing the change in functional groups, and the results clearly reflected the influence of WBER on the cement hydration process and products. The broad peak in the range of 3000–3700 cm^−1^ corresponded to the presence of -OH groups in the hardened WBER-modified cement [46]. Specifically, the peak around 3650 cm^−1^ was attributed to the stretching vibration of -OH groups in calcium hydroxide within the cement hydration products, while the peak at approximately 3430 cm^−1^ was associated with -OH groups in the water of other hydration products. The presence of -OH bonds in all hardened WBER-modified cement samples indicated the formation of well-reacted hydration products across all groups. The experimental group exhibited a lower transmittance value for -OH compared to the control group, suggesting that WBER enhanced the hydration of cement paste and facilitated the formation of Ca(OH)_2_. The characteristic peak of high-intensity CO_3_^2−^ was also detected in the infrared spectrum, and the wavenumbers were in the range of 1430–1420 cm^−1^ and around 875 cm^−1^, respectively. This may be due to the loose microstructure and high porosity of cement with a high W/C ratio, allowing a significant amount of atmospheric CO_2_ to penetrate the cement paste and react with calcium-containing hydration products to form calcium carbonate. WBER, however, filled the cement paste and reduced the microstructure porosity, thereby impeding CO_2_ reaction with the hydration products within the paste. The characteristic peaks of the silicate phase (namely, C–S–H) in cement hydration products are usually around 960 cm^−1^ and 470 cm^−1^ [47]. However, under the condition of a relatively obvious carbonation effect, polymerization of the C–S–H gel chain will be promoted and the wavenumbers of its characteristic peaks will be pushed up [48]. Therefore, the peak with a wavenumber around 1000 cm^−1^ in Figure 10 was the Si–O bond in the C–S–H gel. In Figure 10, the influence of WBER on the Si–O bond was small, which may have been related to the additional amount of WBER. A small amount of WBER could increase the production of C–S–H, whereas an excessive amount may have disrupted normal cement hydration, leading to a decrease in the peak value of the Si–O bond.

#### 3.2.2. X-ray Diffraction (XRD)

The X-ray diffraction pattern results are shown in Figure 11, and the XRD results indicated the influence of WBER on the crystal form of cement hydration products. As depicted in Figure 11, the addition of WBER did not induce a fundamental alteration in the crystal form of the hydration products, and the main hydration products remained as portlandite, calcite, quartz, and other unhydrated mineral clinkers. The early strength of cement is primarily attributed to the formation of C–S–H in hydration products. The main strength in the early stage of cement is derived from the generation of C–S–H in hydration products. Meanwhile, C–S–H gel is poorly crystalline and usually cannot be directly qualitatively analyzed by XRD. Portlandite in the hydration product is a byproduct of the reaction between C_2_S and C_3_S and water in the cement hydration process, and the reaction equation is as follows:(4)$$3CaO\cdot {SiO}_{2}+nH_{2}O\to xCaO\cdot {SiO}_{2}\cdot yH_{2}O+(3-x){Ca(OH)}_{2}-$$
(5)$$2CaO\cdot {SiO}_{2}+nH_{2}O\to xCaO\cdot {SiO}_{2}\cdot yH_{2}O+(2-x){Ca(OH)}_{2}$$

The amount of C–S–H gel generated can be deduced by assessing the reaction involving Ca(OH)_2_ within the hydration byproducts. The diffraction peak of Ca(OH)_2_ is at the 2θ angle range of 17.5–18.5°. Compared with the control group, the diffraction peak of the experimental group rose, and it can be speculated that the appropriate amount of WBER had a promoting effect on the generation of portlandite in cement hydration products and also promoted the hydration reaction of cement. The 2θ angles were in the range of 28.8–30.0° and 26.2–27.0° for the diffraction peaks of CaCO_3_ and SiO_2_, respectively. The addition of WBER significantly decreased the diffraction peaks of CaCO_3_ and SiO_2_ in the hardened modified cement, indicating that the introduction of WBER affected the carbonization process. This may be attributed to WBER filling numerous voids in the hardened cement, thereby improving the pore structure and reducing the entry of carbon dioxide into the cement matrix, which in turn enhances the carbonation resistance and durability of the hardened cement. This finding aligned with the previous FT-IR analysis.

#### 3.2.3. Scanning Electron Microscopy (SEM)

The SEM images of WBER-modified cement grout and the control group under different magnification ratios are shown in Figure 12. At low magnification (Figure 12A,B), the experimental group did not differ much from the control group, and WBER was able to mix evenly with cement, which also had similar compactness after hardening. The effect of polymers on the microstructure of the cement paste can be seen in local images at high magnification. The distribution characteristics of polymers in the composite grouting materials can be observed in Figure 12F, and the existence of micro-cracks, which may decrease the strength and increase the permeability, was observed in the interfacial transition zone between WBER and the cement hydration products. The effect of WBER on the cement hydration products can be seen by the alignment in Figure 12G,H. C–S–H/C–A–S–H is an amorphous, dense gelatinous substance, generally in fibrous or fluffy clusters, which are formed in large quantities around cement particles. The needle-like ettringite (AFt) nucleate grows between the pores of cement particles. Compared with the control, more formation of C–S–H and AFt could be found in Figure 12H, which was consistent with the discussion about C–S–H in the XRD and FT-IR experiments. On the one hand, WBER was beneficial to the hydration reaction of cement, while on the other hand, the formation and growth of hydration reaction products such as C–S–H, CH, and AFt hindered the film formation of polymers. The polymer was wrapped around the cement particles due to electrostatic adsorption and underwent further reaction with cement, and the reaction products filled between the cement particles to fill in the large pores and cracks. But WBER may have entrapped a portion of air when mixed, and although TBP was used for defoaming in this paper, there was still a large number of pores in WBER-modified grout from the contrast in Figure 12G,H, which may have partially compromised the strength.

#### 3.2.4. Confocal Laser Scanning Microscopy (CLSM)

Confocal laser scanning microscopy (CLSM) was introduced into the study of the polymer-modified cement-based materials. CLSM is an optical technique that relies on fluorescence detection to directly observe the adsorption of polymers onto cement particles and to construct high-resolution maps of the spatial distribution of water, cement, and polymer components. The W/C ratio of the fresh suspension was maintained at 0.4, WBER/CA = 2:1, and the total polymer content was 1% (Figure 13) and 10% (Figure 14), respectively. To differentiate the polymer and aqueous phases, lipophilic Nile red and hydrophilic sodium fluorescein dyes were utilized for labeling purposes. For each fresh cement paste sample, a series of three CLSM images was acquired. The image on the left (Figure 13a,d and Figure 14a,d) was captured using an excitation wavelength of 561 nm, and Nile red exhibited strong orange-red fluorescence under excitation at this wavelength, indicating the distribution of the polymer in the suspension. The middle image (Figure 13b,e and Figure 14b,e) was taken at an excitation wavelength of 488 nm, with sodium fluorescein emitting strong green fluorescence to represent the water phase distribution. The rightmost image (Figure 13c,f and Figure 14c,f) is a composite of the previous two, clearly illustrating the spatial distribution of the polymer within the fresh cement grout.

The CLSM image depicting the distribution of 1% polymer in fresh cement paste is presented in Figure 13. The spherical red particles in Figure 13a,d are colloidal particles stained with Nile red dye, dispersed throughout the fresh cement paste, with individual polymer particles readily identifiable. As observed in Figure 13c,f, the red dots are predominantly found in proximity to the cement particles (black in the images), suggesting that upon contact with cement, the polymer particles became fixed within the fresh cement paste and were loosely attached near the cement surface. This phenomenon may be attributed to the electrostatic adsorption resulting from the zeta potential discussed earlier. The opposite charges between the polymer and cement particles generated an adhesive force during mixing. Figure 14 illustrates the CLSM images depicting the distribution of 10% polymer in fresh cement paste. It is seen in Figure 14a,d that the number of colloid particles (in red) stained by Nile red dye increased significantly and the distribution in the paste was denser. Compared with the green in Figure 13c,f, the appearance of yellowish-green in Figure 14c,f indicates that more polymer particles appeared in the aqueous phase of fresh cement paste, while no significant change in green is found in Figure 13, indicating that there was less polymer content in the aqueous phase at this time. The addition of more polymers not only promoted the adsorption between polymer particles and cement particles but also greatly increased the content of the polymer in aqueous suspension. That is, after the polymer content was increased, a large number of polymer particles were adsorbed and fixed to the cement particles, while some of the remaining polymer particles still flowed in the water phase (yellowish-green). The polymer particles in the aqueous phase solidified to form a polymer network structure wrapped in cement particles. It is then reasonable to believe that the distribution of polymer particles in the water phase and the flocculation with cement particles were the reasons for the increase in the stability of the polymer-modified grouting. When the amount of polymer was small, the polymer adsorbed on the surface of cement particles and played a “ball-bearing” effect. When the amount of polymer was too large, the fluidity of the grouting was reduced, and the yield stress and plastic viscosity were increased.

### 3.3. Mechanical Properties of WBER-Modified Composite Grouting

#### 3.3.1. Uniaxial Compressive Strength

Figure 15 presents the uniaxial compressive strength values of WBER-modified cement paste, varying the ratios of WBER to CA. Compared with the control group, the uniaxial compressive strength of the four experimental groups was reduced only in the non-CA group, and the 3 d uniaxial compressive strength was 3.86 MPa, which was reduced by 30.76% compared with 5.56 MPa for the control group. The uniaxial compressive strength was 8.16 MPa at 28 d, which was 10% lower than that of the control group (9.08 MPa). It can be seen that the uniaxial compressive strength of the cement was adversely affected by pure WBER without CA, and the strength was weak in the early stage but developed rapidly in the later stage. Compared to the experimental groups with different CA dosages, it can be seen that the compressive strength of WBER-modified cement increased first and then decreased with the increase in the proportion of CA in the modifier. The optimum strength ratio of WBER to CA was about 2:1, and the compressive strength of the hardened cement was 12.61 MPa at this time, which was 38.9% higher than that of the neat cement paste.

#### 3.3.2. Splitting Tensile Strength

Figure 16 shows the influence of the ratio of WBER to CA on the splitting tensile strength of hardened WBER-modified cement paste. It can be seen from Figure 16 that WBER enhanced the splitting tensile strength of the composite cement, and the strength of all groups with WBER was higher than that of the neat cement. The splitting tensile strength of the composite cement increased first and then decreased with an increasing proportion of CA in the modifier. The highest strength ratio was WBER/CA = 2:1. The strength values at 7 d and 28 d were 0.86 MPa and 1.16 MPa, respectively, which were 56.4% and 48.7% higher than those of the control group at 0.55 MPa and 0.78 MPa, respectively.

The mechanism underlying the impact of WBER on the mechanical properties of cement can be expounded through two distinct pathways. Firstly, the findings from the FT-IR and XRD analyses suggested that WBER enhances the hydration of cement particles, leading to a substantial increase in the production of C–S–H and Ca(OH)_2_, which are crucial for the early strength of cement. However, the formation of a network membrane structure by WBER within the cement matrix can impede further hydration and, consequently, hinder the development of mechanical strength to some degree. Moreover, the SEM images indicated the potential presence of micro-cracks in the interfacial transition zone between WBER and cement particles, which could adversely affect the strength development of the modified grout.

Secondly, the zeta potential of WBER is opposite to that of cement particles. Within the suspension system, WBER is electrostatically adsorbed onto the surface of cement particles, thereby enhancing the cohesion of the suspension post-hardening. The smaller particle size of WBER allows it to fill the micro-pores between cement particles, optimizing the internal pore distribution of the grout and contributing to improved mechanical strength.

### 3.4. The Modification Mechanism of WBER-Modified Cement

Through the above physical property tests, rheological property tests, and characterization analysis, the mechanism of action of WBER-modified cement paste can be deduced. According to various studies, functional groups such as -OH and -COOH within polymers like polyacrylate latex (PA) [49], carboxylic styrene-butadiene latex (XSBRI) [50], and styrene-butadiene latex (SBR) [51] are electrostatically adsorbed onto the polymer chain. This adsorption occurs due to the membrane structure’s charge distribution. Subsequently, the -OH and -COOH groups in the polymer react with Ca^2+^ ions, leading to the formation of complex hydration products. According to the above experiments and similar literature review [52], the addition of WBER to cement paste does not produce new reaction products, but the main reaction is the reaction of -OH in WBER with Ca^2+^ produced by the hydrolysis of cement.

The reaction process is shown in Figure 17. In accordance with the chemical reaction principle in Figure 2, WBER reacts with CA to form a polymer chain, and the epoxy group opens to form the hydroxyl functional group. As illustrated in Figure 17a, after cement particles and epoxy polymer chains are evenly mixed in water, WBER also hydrolyzes under alkaline conditions (cement becomes alkaline after undergoing a hydrolysis reaction).The hydroxyl functional groups within WBER react with hydroxide ions in an aqueous medium to remove hydrogen ions and generate water. Concurrently, the partial hydrolysis of cement particles releases a significant number of calcium ions into the aqueous medium. From Figure 17b,c, it is seen that there are two kinds of reactions between the hydration products and polymer chains: one is two epoxy polymer chains connected by the reaction of calcium ions, and the other is the polymer chain connected by calcium ions to silicon dioxide on the cement surface. There are large amounts of -OH and Ca^2+^ in the solution, and the polymer generates electrostatic attraction due to its opposite zeta potential to cement particles.

The crosslinking of numerous epoxy chains forms a polymer network that fills the voids within the cement matrix, with Ca^2+^ ions serving as the crosslinking points. The CLSM results further proved that the surface of colloidal polymer particles binds to the surface of cement particles through electrostatic adsorption, the reaction products are generated through the subsequent reaction of hydroxyl and Ca^2+^, and then cement particles and cement particles are bridged to form flocculating structures. This crosslinked epoxy resin network is integrated with the cement hydration products, establishing a robust network that connects the organic and inorganic components within the polymer-modified cement system.

## 4. Conclusions

This study evaluated the potential of using WBER as a modifier for cement-based grouting materials. The focus is on investigating the effects of the proportions of WBER and CA on the rheological properties, mechanical properties, and chemical characteristics of composite grouts. Based on the experimental findings, the following conclusions are drawn:WBER-modified grout exhibits Bingham fluid characteristics, with WBER enhancing the fluidity of cement paste and prolonging its stability after water addition, effectively mitigating rapid fluidity loss.The addition of WBER modifies the particle size distribution of the cement paste, resulting in a decrease in mean particle size and a more uniform grain size distribution. The contrasting zeta potentials between WBER and the cement particles lead to an initial increase in the suspension’s zeta potential, which diminishes as the CA content rises.Incorporation of WBER alone diminishes the uniaxial compressive strength of the hardened grout. However, the composite’s strength is significantly improved with the addition of CA, reaching an optimal WBER/CA ratio of approximately 2:1, corresponding to a 38.88% strength enhancement. The splitting tensile strength is markedly affected by WBER, with the optimal epoxy-to-curing agent ratio of 2:1 yielding a 48.7% increase.WBER promotes the cement hydration process, leading to an augmentation in the yield of calcium hydroxide within the hydration products, while concurrently diminishing the concentration of calcium carbonate. The core reaction between WBER and cement is the reaction between free Ca^2+^ and -OH, with Ca^2+^ as the node to form a complex network structure as a product of crosslinking. The reaction products adhere to cement particles, thereby establishing a robust composite of organic and inorganic phases.

## Figures and Tables

**Figure 1 polymers-16-02665-f001:**
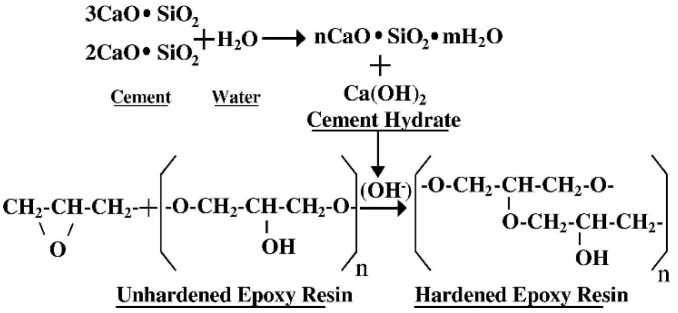
Hardening mechanism of ER without hardener in cement paste [38].

**Figure 2 polymers-16-02665-f002:**
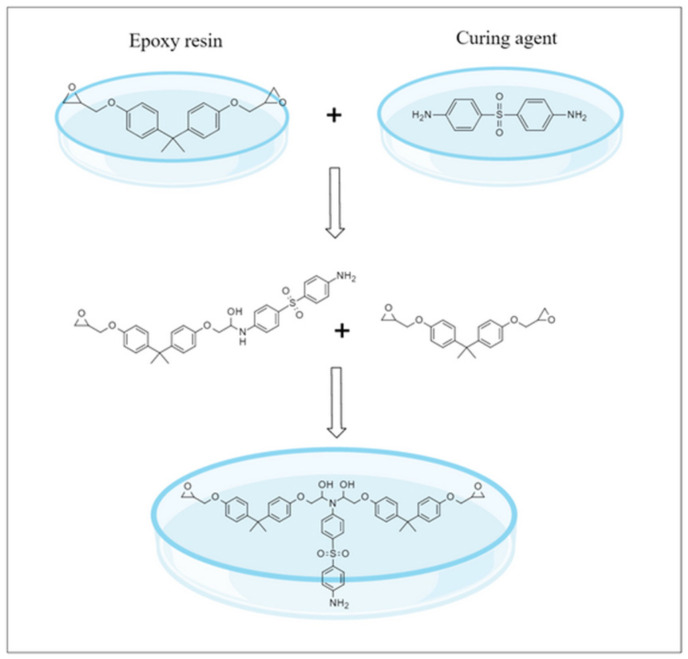
Curing reaction of WBER with CA [30].

**Figure 3 polymers-16-02665-f003:**
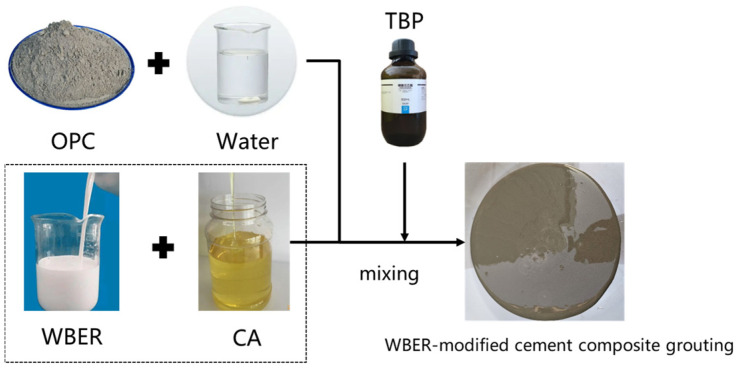
Preparation of WBER-modified cement-based paste.

**Figure 4 polymers-16-02665-f004:**
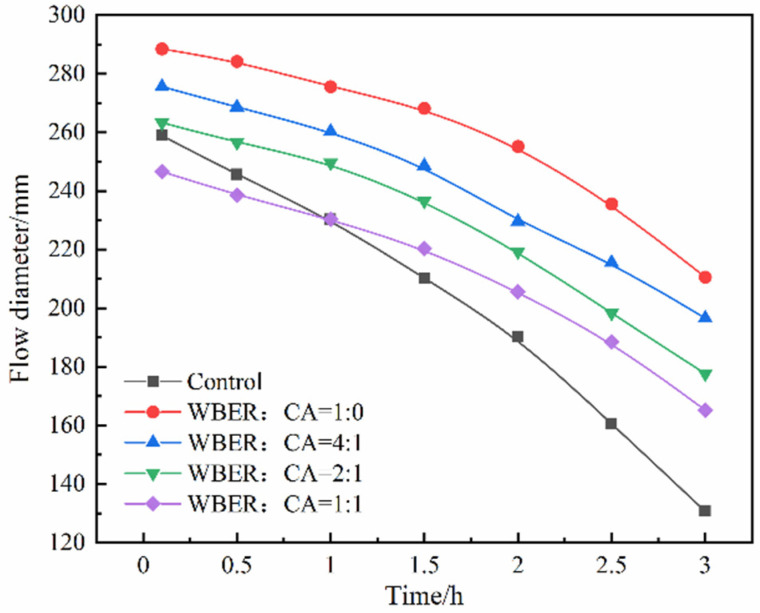
Effect of the ratio of WBER to CA on the fluidity of the composite grout.

**Figure 5 polymers-16-02665-f005:**
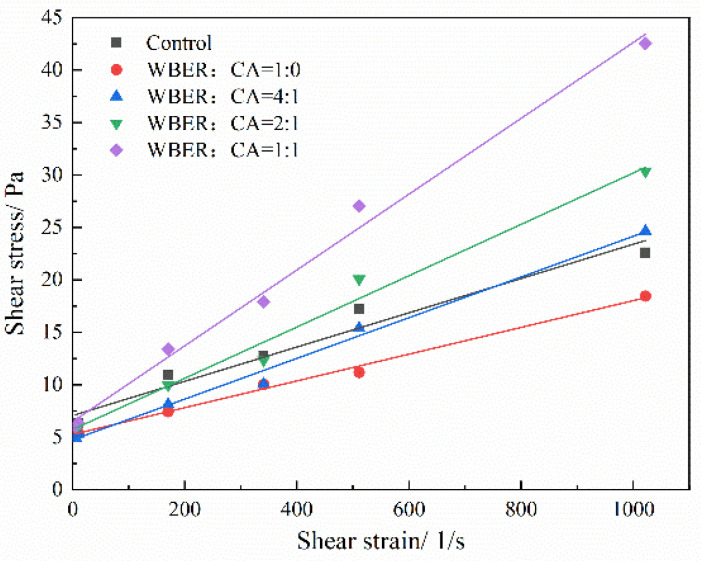
Rheological and fitting curves of WBER-modified composite grouting.

**Figure 6 polymers-16-02665-f006:**
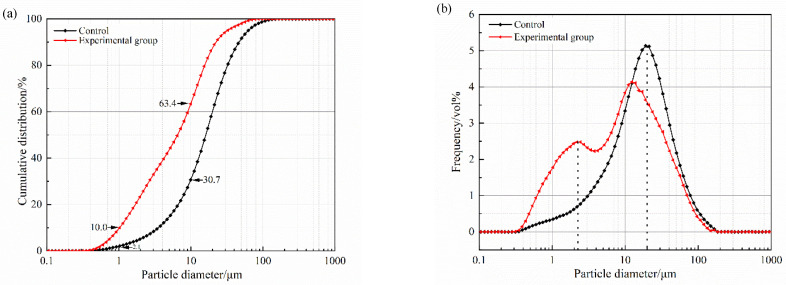
Grain size distribution of WBER-modified grout: (**a**) Cumulation; (**b**) Frequency.

**Figure 7 polymers-16-02665-f007:**
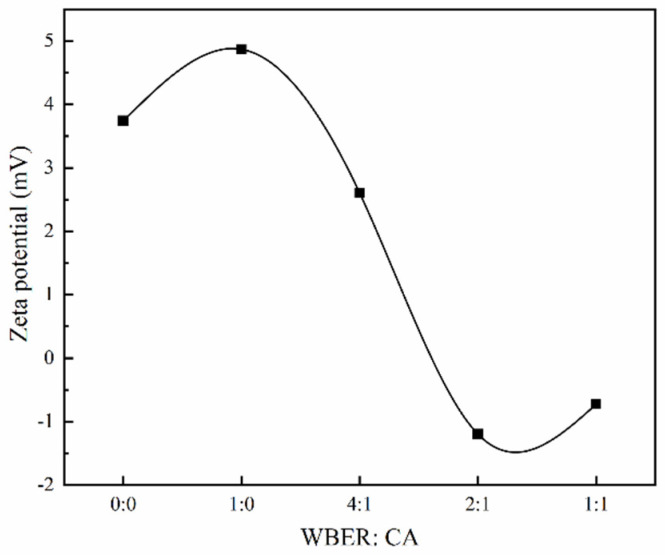
Zeta potential of WBER–modified composite paste.

**Figure 8 polymers-16-02665-f008:**
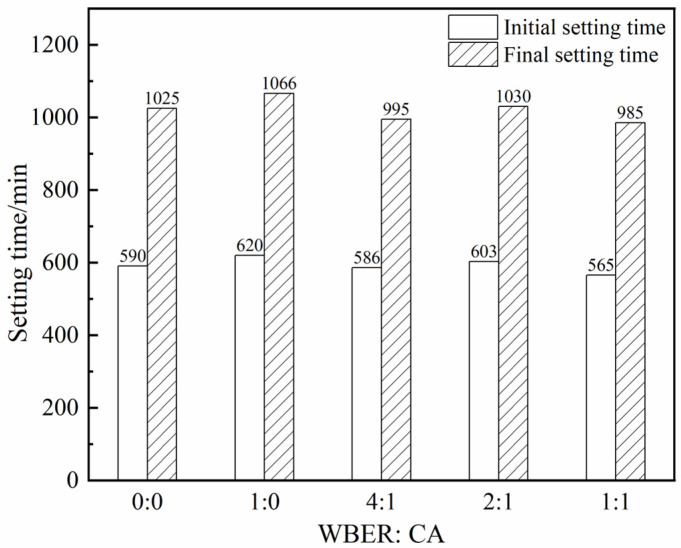
Setting time of WBER-modified composite grout.

**Figure 9 polymers-16-02665-f009:**
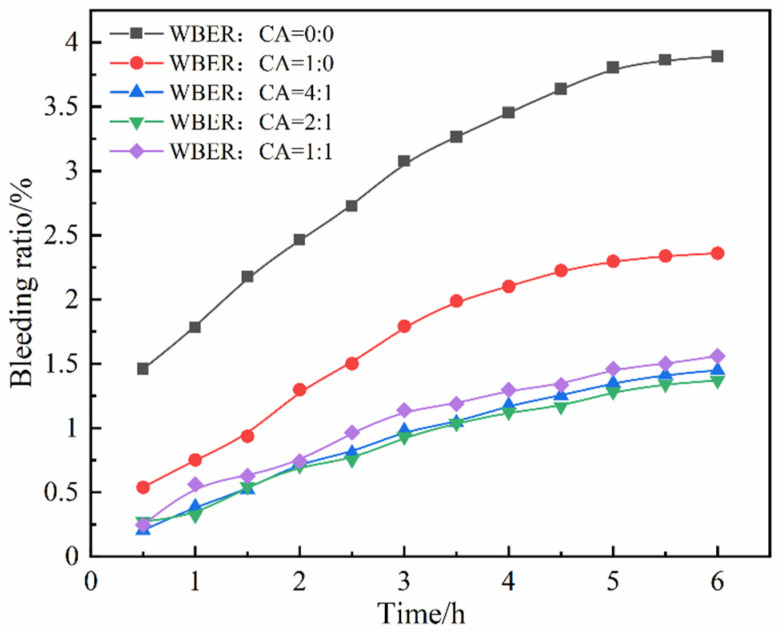
Bleeding ratios of WBER-modified composite paste.

**Figure 10 polymers-16-02665-f010:**
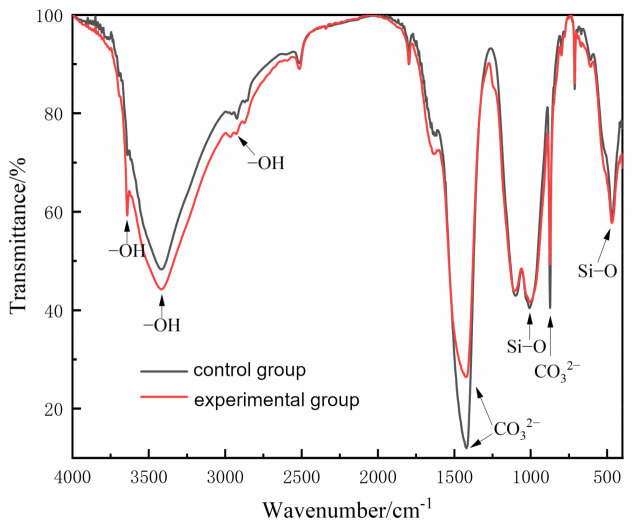
FT-IR spectra of WBER-modified composite paste.

**Figure 11 polymers-16-02665-f011:**
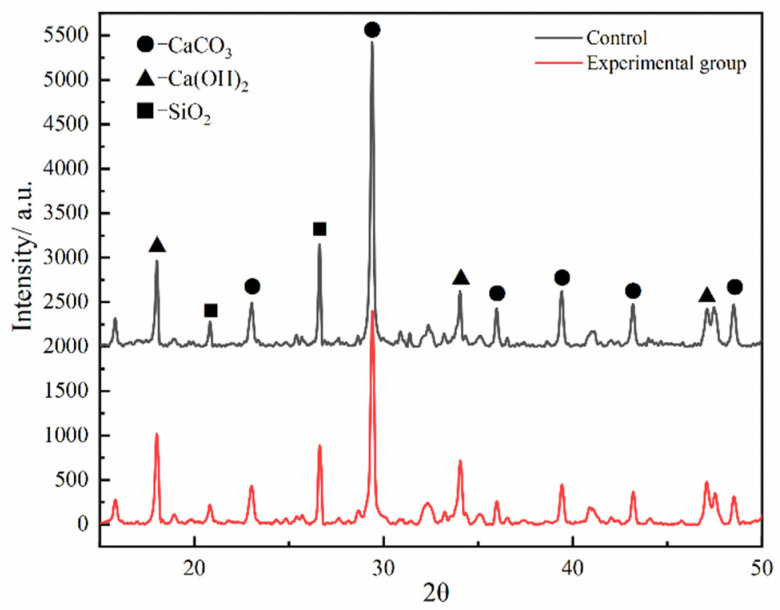
XRD pattern of WBER-modified composite paste.

**Figure 12 polymers-16-02665-f012:**
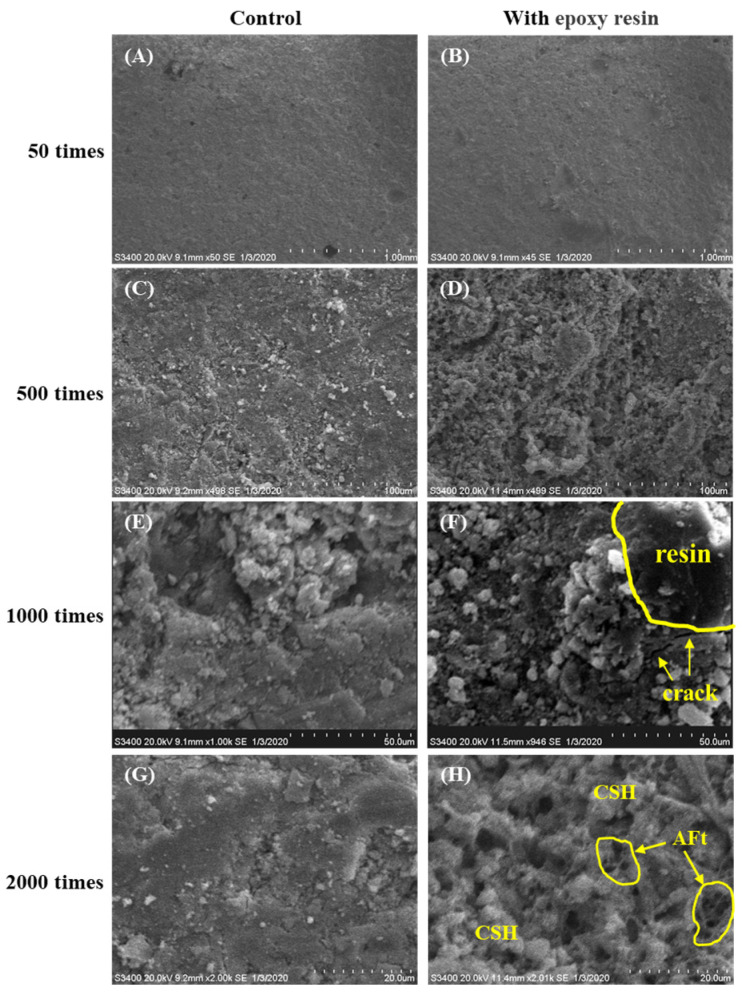
SEM images of WBER-modified composite paste. ((**A**,**B**): 50 times; (**C**,**D**): 500 times; (**E**,**F**): 1000 times; (**G**,**H**): 2000 times).

**Figure 13 polymers-16-02665-f013:**
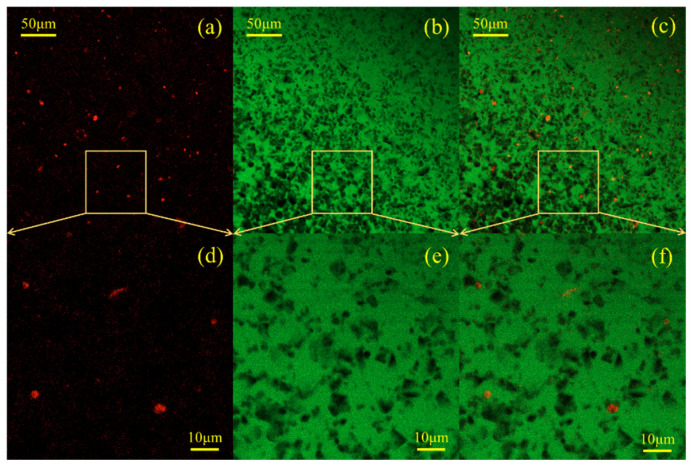
CLSM images of WBER-modified cement paste with the addition of 1% polymer. (**a**,**d**): fluorescence in polymer phase; (**b**,**e**): fluorescence in water phase; (**c**,**f**): overlapped fluorescence images of the water phase and polymer phase.

**Figure 14 polymers-16-02665-f014:**
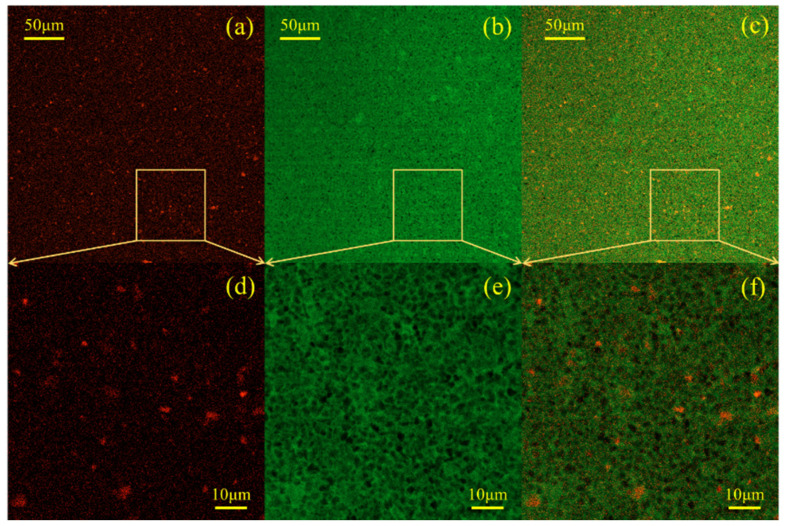
CLSM images of WBER-modified cement paste with the addition of 10% polymer. (**a**,**d**): fluorescence in polymer phase; (**b**,**e**): fluorescence in water phase; (**c**,**f**): overlapped fluorescence images of the water phase and polymer phase.

**Figure 15 polymers-16-02665-f015:**
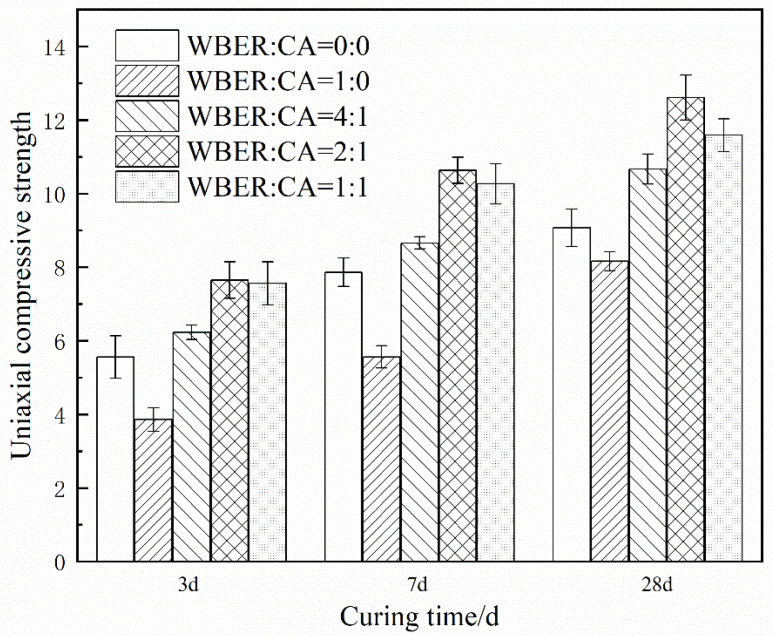
Uniaxial compressive strength of hardened WBER-modified cement paste.

**Figure 16 polymers-16-02665-f016:**
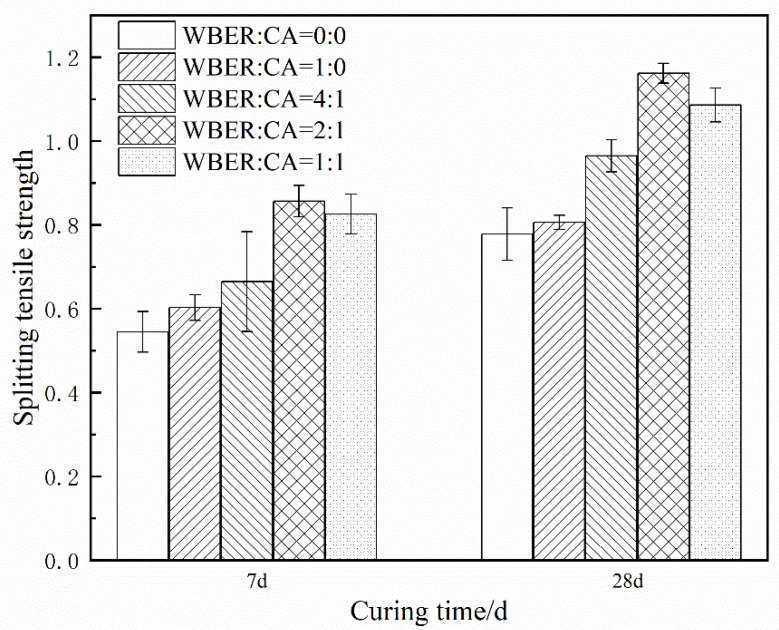
Splitting tensile strength of hardened WBER-modified cement paste.

**Figure 17 polymers-16-02665-f017:**
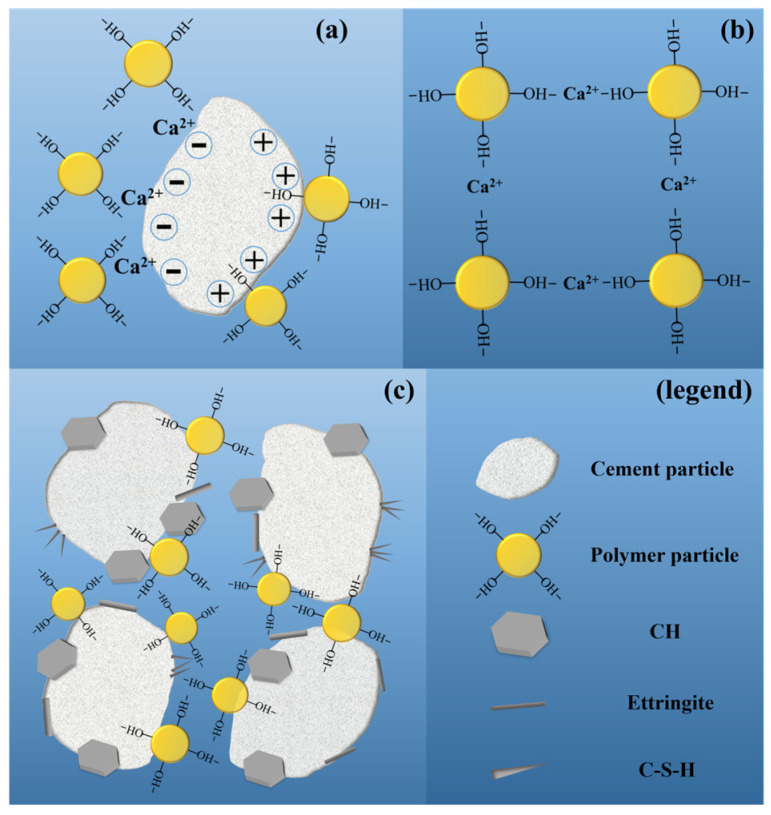
Schematic representation of crosslinking reaction process between epoxy resin and ionized calcium. ((**a**): hydrolysis of polymers and cement; (**b**): the reaction between Ca^2+^ and hydroxyl groups; (**c**): reaction between polymer and cement).

**Table 1 polymers-16-02665-t001:** Properties of WBER and CA.

Item	Epoxy Resin	Curing Agent
Appearance	Milky white	Light yellow
Solid content	50 ± 2	44 ± 2
Epoxide (active hydrogen) equivalent	220~280	260 ± 60
Viscosity mpa·s/24 °C	100~450	>2000
pH value	2~7	8~11
Specific gravity	1.01~1.08	1.00~1.08

**Table 2 polymers-16-02665-t002:** Properties of defoamer.

Molecular Formula	Molecular Weight	Melting Point	Boiling Point	Water Solubility	Refractive Index	Flash Point	Density
C_12_H_27_O_4_P	266.31	−79 °C	289 °C	0.6 g/100 mL	1.423~1.425	146 °C	0.979

**Table 3 polymers-16-02665-t003:** The Bingham model fitting parameters of epoxy-modified paste.

WBER/CA	Yield Stress (*τ*_0_, Pa)	Plastic Viscosity (*η,* Pa·s)	Correlation Coefficient (*R*^2^)
0:0	6.60	17.70	0.99
1:0	5.61	12.74	0.99
4:1	4.68	18.94	0.99
2:1	5.55	24.01	0.99
1:1	6.48	35.68	0.99

## Data Availability

The original contributions presented in the study are included in the article, further inquiries can be directed to the corresponding author.

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
