# Peer review of "Effects of Polymer–Curing Agent Ratio on Rheological, Mechanical Properties and Chemical Characterization of Epoxy-Modified Cement Composite Grouting Materials"

_polymers, 2024, doi:10.3390/polym16182665_

Round 1
Reviewer 1 Report
Comments and Suggestions for Authors
1. Title: it is better to clearly define the material type, such as cement-based composite grouting material.
2. Abstract: The fabrication method is missing. There are many findings from experimental works, and please select the main findings with 3 points.
3. The last paragraph of the Introduction section does not clearly discuss on research scope, in addition, the aim of this study is not summarized in 1 paragraph.
4. 2.2 section, it is suggested to draw the flowchart of the preparation process.
5. 2.3 Experimental approaches: For the introduced equations, please cite some fundamental refs.
6. Results and discussion section: Figure 3 what is the control line? Is it the pure composite grout with WBER: CA=0: 0? If possible, please clearly define it.
7. Page 7 the yield stress highlights the initial flow of grout, please cite refs to support the findings.
8. Ca(OH)2 changes to Ca(OH)2, please check the whole manuscript.
7. For the mechanical properties of WBER-modified composite grouting, the data is required to add an error bar, which can provide significant findings.
8. Conclusion: it is ok.
9. Refs is ok.
10. The English of this manuscript is required to be proofread with minor revisions.
Comments on the Quality of English LanguageThe English of this manuscript is required to be proofread with minor revisions.
Author Response
Comments 1: Title: it is better to clearly define the material type, such as cement-based composite grouting material.
Response 1: Thank you for your suggestion on the title. We have included "cement" in the title, and the new title is (Effects of polymer-curing agent ratio on rheological, mechanical properties and chemical characterization of epoxy-modified cement composite grouting materials).
Comments 2: Abstract: The fabrication method is missing. There are many findings from experimental works, and please select the main findings with 3 points.
Response 2: We are very grateful for your suggestion. We have made significant revisions to the abstract section, refining the language and adjusting it to three key points.
New abstract: ” This study designs and uses water-borne epoxy resin (WBER) and curing agent (CA) to modify traditional cement-based grouting for the tunnel. The purpose of this paper is to analyze the rheological and mechanical properties of composite grouting with different ratios of WBER and CA, and analyze the modification mechanism by means of chemical characterization, to explore the feasibility of WBER as a high-performance modifier for tunnel construction. The composite grouting is prepared by mixing cement paste with polymer emulsion. A series of experiments were carried out to investigate the effects of WBER and CA on slump test, viscosity, rheological curve, setting time, bleeding rate, grain size distribution, zeta potential, compressive and splitting tensile strength, X-ray diffraction(XRD), Fourier transform infrared spectrometer(FTIR), and scanning electron microscope (SEM) of the composite grout. The results show that WBER improves grout fluidity, which decreases in combination with CA, while also reducing the average particle size of the composite grout for a more rational size distribution; Optimal uniaxial (38.9%) and splitting tensile strength (48.7%) of the grout are achieved with a WBER to CA mass ratio of 2:1; WBER accelerates cement hydration, with the modification centered on the reaction between free Ca2+ and polymer -OH, significantly enhancing the strength, fluidity, and stability of the polymer-modified composite grout compared to traditional cement-based grouting.”
Comments 3: The last paragraph of the Introduction section does not clearly discuss on research scope, in addition, the aim of this study is not summarized in 1 paragraph.
Response 3: Thank you for your suggestion. We have incorporated a summary of the research purpose at the end of the introduction to ensure the article more closely aligns with the objectives of the study.
New last paragraph of the Introduction section:” Although WBER-modified cement mortar has been studied and applied in civil engineering, its application in grouting engineering has not been well explored. With the rapid development of deep underground engineering, the formation, prediction, and control of geological disasters such as fracture, swelling, deformation, and permeability of deep surrounding rock are becoming increasingly prominent. Traditional cement-based grouting materials have poor bonding performance and brittle failure characteristics, which make it difficult to meet the reinforcement needs of nonlinear large deformation and large block displacement. Therefore, a new ER-modified composite grouting was prepared using cement paste and polymer liquid composed of WBER, CA, and defoamer with different ratios. The rheological and mechanical properties of composite grouting were studied through a series of experiments, including fluidity, setting time, bleeding, viscosity, and rheological curve, grain size distribution and zeta potential, uniaxial compressive strength, and splitting tensile strength. The composition, crystal form, and functional groups of the hydration products of grouting were studied by XRD and FT-IR. The micro-morphology of composite grouting was observed by SEM, and the modification mechanism of WBER to neat cement paste was discussed.”
Comments 4: 2.2 section, it is suggested to draw the flowchart of the preparation process.
Response 4: Added Figure 3. Preparation of WBER-modified cement-based paste
Comments 5: 2.3 Experimental approaches: For the introduced equations, please cite some fundamental refs.
Response 5: Thank you for your suggestions. The equations that require explanation in the text are in the 2.3.1 section, namely equations 1 and 2, which are used to calculate the shear stress and apparent viscosity. These equations are taken directly from the instrument’s manual, and therefore, no references are provided.
Comments 6: Results and discussion section: Figure 3 what is the control line? Is it the pure composite grout with WBER: CA=0: 0? If possible, please clearly define it.
Response 6: Your opinion is very correct. In this article, the control group is pure cement paste (WBER: CA=0: 0). We have added the relevant description in Section 2.2 of the text (in red).
Comments 7: Page 7 the yield stress highlights the initial flow of grout, please cite refs to support the findings.
Response 7: Thank you for your suggestion. Literature references have been included at the appropriate locations in the text, as per your suggestion. “The yield stress is an important index to measure the initial flow of grout, and the flow of paste must satisfy the condition that shear stress exceeds its yield stress [29]”.
Comments 8: Ca(OH)2 changes to Ca(OH)2, please check the whole manuscript.
Response 8: I am very sorry for this oversight. We have carefully checked the spelling of calcium hydroxide throughout the entire text and made corrections.
Comments 9:For the mechanical properties of WBER-modified composite grouting, the data is required to add an error bar, which can provide significant findings.
Response 9: Thank you for your suggestion. We added error bars in the UCS and TST figures.
Comments 10:Conclusion: it is ok.
Response 10: Thank you
Comments 11: Refs is ok.
Response 11: Thank you
Comments 12. The English of this manuscript is required to be proofread with minor revisions.
Response 12: Thank you for your suggestion. We have proofread the article as a whole and modified some sentences to make them more in line with English expression.

Reviewer 2 Report
Comments and Suggestions for Authors
Based on the overall quality, a major revision is needed.
Comments:
(1) Why did you choose WBER and CA to modify the grouts?
(2) The grouting materials you investigated is used for backfilling of shield tunnel or mountain tunnel?
(3) What is the research gap? Please highlight your contribution and novelty in the introduction.
(4) More recent studies about grouting materials for tunnels should be discussed and added, such as "A state-of-art review on development and progress of backfill grouting materials for shield tunneling"ï¼›“Nauclea latifolia herb root waste reinforced epoxy polymer composite: The study of effects, modelling, certainty and sensitivity analysis.”
(5) Figure 6. What do you mean by this figure?
(6) Figure 7 is not clear.
(7) Figure 10, please revise this figure. There are some mistakes in the peaks' phases.
(8) Figure 11 is not clear.
(9) Figure 16. Please redraw this one.
(10) Conclusions. Please make it concise. In addition, one general law should be concluded at the beginning of each point.
Comments on the Quality of English LanguageThe language should be polished.
Author Response
Comments 1:Why did you choose WBER and CA to modify the grouts?
Response 1: Thank you for your question. With the rapid development of deep underground engineering, the formation, prediction, and control of geological disasters such as fracture, swelling, deformation, and permeability of deep surrounding rock are becoming increasingly prominent. Traditional cement-based grouting materials have poor bonding performance and brittle failure characteristics, which make it difficult to meet the reinforcement needs of nonlinear large deformation and large block displacement. The reason we chose waterborne epoxy resin is its good physical and mechanical properties, strong bonding ability, and its characteristic of being able to cure in an alkaline environment, which is very compatible with the cement paste system.
Comments 2: The grouting materials you investigated is used for backfilling of shield tunnel or mountain tunnel?
Response 2: Thank you for your question. The grouting materials we investigated is used for backfilling of the shield tunnel.
Comments 3:What is the research gap? Please highlight your contribution and novelty in the introduction.
Response 3: Thank you for your question. Our previous research has found that the curing agent has a significant impact on the properties of epoxy resin-modified cement composites. At the same time, the use of the curing agent is more complex than that of pure resin, and there is theoretical support indicating that epoxy resin can self-cure under alkaline conditions. Therefore, this paper aims to explore the role of the curing agent in cement epoxy resin-modified cement materials through experiments.
Comments 4:More recent studies about grouting materials for tunnels should be discussed and added, such as "A state-of-art review on development and progress of backfill grouting materials for shield tunneling"ï¼›“Nauclea latifolia herb root waste reinforced epoxy polymer composite: The study of effects, modelling, certainty and sensitivity analysis.”
Response 4: Thank you for your comment. We have carefully read the two articles you recommended, which are relevant to our study, and have included them in the references with numbers 1 and 34.
Comments 5:Figure 6. What do you mean by this figure?
Response 5: Thank you for your comment. This paper uses the zeta potential test to measure the surface charge of particles in the composite slurry and then analyzes the dispersion and movement behavior of the particles in the paste. Zeta potential represents the surface charge state of particles. In the study of grouting fluidity, the zeta potential on the particle surface determines the strength of the electrostatic electric-dominated interaction between particles, which directly determines the rheological energy of suspension. The greater the absolute value of zeta potential, the stronger the dispersion between solution particles.
Comments 6: Figure 7 is not clear.
Response 6: Thank you for your comment. The issue of unclear images may be due to Word automatically compressing the image quality. We have re-uploaded the images, and if there are still problems, I can provide the original images in TIF format.
Comments 7: Figure 10, please revise this figure. There are some mistakes in the peaks' phases.
Response 7: We have checked the annotation of the diffraction peaks on this image and compared it with the standard spectrum, and found no issues. Could you please further specify where the error is? Thank you very much.
Comments 8: Figure 11 is not clear.
Response 8: Thank you for your comment. The issue of unclear images may be due to Word automatically compressing the image quality. We have re-uploaded the images, and if there are still problems, I can provide the original images.
Comments 9: Figure 16. Please redraw this one.
Response 9: Thank you for your comment. Could you please specify which parts of this figure are not clear enough and need to be redrawn? Since the time for modification is tight, if it is indeed necessary to redraw, I will need some time to complete it.
Comments 10: Conclusions. Please make it concise. In addition, one general law should be concluded at the beginning of each point.
Response 10: Thank you for your comment. Based on your suggestion, we have revised the conclusion section of the paper.
- “WBER-modified grout exhibits Bingham fluid characteristics, with WBER enhancing the fluidity of cement paste and prolonging its stability after water addition, effectively mitigating rapid fluidity loss.
- The addition of WBER modifies the particle size distribution of the cement paste, resulting in a decrease in mean particle size and a more uniform grain size distribution. The contrasting zeta potentials between WBER and cement particles lead to an initial increase in the suspension’s zeta potential, which diminishes as the CA content rises.
- Incorporation of WBER alone diminishes the uniaxial compressive strength of the hardened grout. However, the composite’s strength is significantly improved with the addition of CA, reaching an optimal WBER: CA ratio of approximately 2:1, corresponding to a 38.88% strength enhancement. The splitting tensile strength is markedly affected by WBER, with the optimal epoxy-to-curing agent ratio of 2:1 yielding a 48.7% increase.”

Reviewer 3 Report
Comments and Suggestions for Authors
After reading the work, I noted for myself a few remarks, but they do not reduce the overall positive impression and the work can be accepted for publication after minor revision.
Abbreviations when used for the first time should be deciphered.
Section 3.1.2 When explaining the influence of the WBER : CA ratio on rheological properties, in my opinion you do not take into account the kinetics of the curing process of epoxy matrices. It is known that in the process of curing of epoxy resin by amines, heat is generated (as a result of which the viscosity of the resin decreases), and the first stage of curing is the formation of gel, which significantly reduces the viscosity of the system. In your case you need to pay attention to the amount of hardener, for your hardener with 4 amine groups under normal conditions the ratio varies from about 10 to 15 wt%. In your work the amount of hardener is much higher and reaches the value of 1 to 1, which should and has an effect on the rheological properties of the mixture probably due to accelerated gelation.
https://www.mdpi.com/2079-4991/14/7/602 In your work you can pay attention to curing curves of epoxy resin when curing with similar hardener. In this case, the curing process is recorded in relation to the heat released as a result of the chemical reaction.
Figure 9. Decipher Control and Experimental group.
‘This may be attributed to WBER filling numerous voids in the hardened cement, thereby improving the pore structure and reducing the entry of carbon dioxide into the cement matrix, which in turn enhances the carbonation resistance and durability of the hardened cement. This finding aligns with the previous FT-IR analysis.’ - Is it possible to make such a statement based on XRD analysis. SEM section is more suitable to confirm this thesis.
Author Response
Comments 1: Abbreviations when used for the first time should be deciphered.
Response 1: Thank you for your comment. The issue of using Abbreviations without clarification for the first time mainly occurred in the abstract, and it has been annotated in the abstract.
Comments 2: Section 3.1.2 When explaining the influence of the WBER : CA ratio on rheological properties, in my opinion you do not take into account the kinetics of the curing process of epoxy matrices. It is known that in the process of curing of epoxy resin by amines, heat is generated (as a result of which the viscosity of the resin decreases), and the first stage of curing is the formation of gel, which significantly reduces the viscosity of the system. In your case you need to pay attention to the amount of hardener, for your hardener with 4 amine groups under normal conditions the ratio varies from about 10 to 15 wt%. In your work the amount of hardener is much higher and reaches the value of 1 to 1, which should and has an effect on the rheological properties of the mixture probably due to accelerated gelation.
https://www.mdpi.com/2079-4991/14/7/602 In your work you can pay attention to curing curves of epoxy resin when curing with similar hardener. In this case, the curing process is recorded in relation to the heat released as a result of the chemical reaction.
Figure 9. Decipher Control and Experimental group.
Response 2: Thank you very much for your suggestion, which is very enlightening. Regarding the heat release phenomenon of the curing agent, it may interfere with the hydration reaction of cement in the cement-polymer system, thereby affecting the material’s properties. Currently, we have not explored this issue, and it is undoubtedly a research direction worth exploring. Thank you again for your suggestion.
Comments 3:‘This may be attributed to WBER filling numerous voids in the hardened cement, thereby improving the pore structure and reducing the entry of carbon dioxide into the cement matrix, which in turn enhances the carbonation resistance and durability of the hardened cement. This finding aligns with the previous FT-IR analysis.’ - Is it possible to make such a statement based on XRD analysis. SEM section is more suitable to confirm this thesis.
Response 3: Thank you for your question. XRD and FTIR are commonly used as characterization methods for important information such as material crystal forms and functional groups. Here, the aim is to connect the hydration reaction of cement with the macroscopic voids, and a likely cause is proposed, which I personally believe has a high probability. Similar views also appear in (https://doi.org/10.1016/j.conbuildmat.2021.123877).

Round 2
Reviewer 2 Report
Comments and Suggestions for Authors
Thanks for your improvement.